# Rectangular Bounding Process

**Xuhui Fan**
School of Mathematics & Statistics
University of New South Wales
xuhui.fan@unsw.edu.au

**Bin Li**
School of Computer Science
Fudan University
libin@fudan.edu.cn

**Scott A. Sisson**
School of Mathematics & Statistics
University of New South Wales
scott.sisson@unsw.edu.au

## Abstract

Stochastic partition models divide a multi-dimensional space into a number of
rectangular regions, such that the data within each region exhibit certain types of
homogeneity. Due to the nature of their partition strategy, existing partition models
may create many unnecessary divisions in sparse regions when trying to describe
data in dense regions. To avoid this problem we introduce a new parsimonious
partition model – the Rectangular Bounding Process (RBP) – to efficiently partition
multi-dimensional spaces, by employing a bounding strategy to enclose data points
within rectangular bounding boxes. Unlike existing approaches, the RBP possesses
several attractive theoretical properties that make it a powerful nonparametric
partition prior on a hypercube. In particular, the RBP is self-consistent and as such
can be directly extended from a finite hypercube to infinite (unbounded) space. We
apply the RBP to regression trees and relational models as a flexible partition prior.
The experimental results validate the merit of the RBP in rich yet parsimonious
expressiveness compared to the state-of-the-art methods.

## 1 Introduction

Stochastic partition processes on a product space have found many real-world applications, such
as regression trees [5, 18, 22], relational modeling [17, 2, 21], and community detection [26, 16].
By tailoring a multi-dimensional space (or multi-dimensional array) into a number of rectangular
regions, the partition model can fit data using these "blocks" such that the data within each block
exhibit certain types of homogeneity. As one can choose an arbitrarily fine resolution of partition, the
data can be fitted reasonably well.

The cost of finer data fitness is that the partition model may induce unnecessary dissections in sparse
regions. Compared to the regular-grid partition process [17], the Mondrian process (MP) [32] is
more parsimonious for data fitting due to a hierarchical partition strategy; however, the strategy of
recursively cutting the space still cannot largely avoid unnecessary dissections in sparse regions.
Consider e.g. a regression tree on a multi-dimensional feature space: as data usually lie in some local
regions of the entire space, a "regular-grid" or "hierarchical" (i.e. $k$d-tree) partition model would
inevitably produce too many cuts in regions where data points rarely locate when it tries to fit data
in dense regions (see illustration in the left panel of Figure 1). It is accordingly challenging for a
partition process to balance *fitness* and *parsimony*.

Instead of this cutting-based strategy, we propose a bounding-based partition process – the Rectangular
Bounding Process (RBP) – to alleviate the above limitation. The RBP generates rectangular bounding

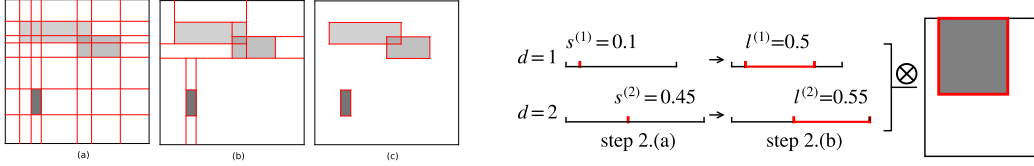

Figure 1: [Left] (a) Regular-grid partition; (b) hierarchical partition; (c) RBP-based partition. [Right] Generation of a bounding box (Step (2) of the generative process for the RBP).

boxes to enclose data points in a multi-dimensional space. In this way, significant regions of the space can be comprehensively modelled. Each bounding box can be efficiently constructed by an outer product of a number of step functions, each of which is defined on a dimension of the feature space, with a segment of value "1" in a particular interval and "0" otherwise. As bounding boxes are independently generated, the layout of a full partition can be quite flexible, allowing for a simple description of those regions with complicated patterns. As a result, the RBP is able to use fewer blocks (thereby providing a more parsimonious expression of the model) than those cutting-based partition models while achieving a similar modelling capability.

The RBP has several favourable properties that make it a powerful nonparametric partition prior: (1) Given a budget parameter and the domain size, the expected total volume of the generated bounding boxes is a constant. This is helpful to set the process hyperparameters to prefer few large bounding boxes or many small bounding boxes. (2) Each individual data point in the domain has equal probability to be covered by a bounding box. This gives an equal tendency towards all possible configurations of the data. (3) The process is self-consistent in the sense that, when restricting a domain to its sub-domain, the probability of the resulting partition on the sub-domain is the same as the probability of generating the same partition on the sub-domain directly. This important property enables the RBP to be extendable from a hypercube to the infinite multi-dimensional space according to the Kolmogorov extension theorem [6]. This is extremely useful for the domain changing problem setting, such as regression problems over streaming data.

The RBP can be a beneficial tool in many applications. In this paper we specifically investigate (1) regression trees, where bounding boxes can be viewed as local regions of certain labels, and (2) relational modelling, where bounding boxes can be viewed as communities in a complex network. We develop practical and efficient MCMC methods to infer the model structures. The experimental results on a number of synthetic and real-world data sets demonstrate that the RBP can achieve parsimonious partitions with competitive performance compared to the state-of-the-art methods.

## 2 Related Work

### 2.1 Stochastic Partition Processes

Stochastic partition processes divide a multi-dimensional space (for continuous data) or a multi-dimensional array (for discrete data) into blocks such that the data within each region exhibit certain types of homogeneity. In terms of partitioning strategy, state-of-the-art stochastic partition processes can be roughly categorized into regular-grid partitions and flexible axis-aligned partitions (Figure 1).

A regular-grid stochastic partition process is constituted by $D$ separate partition processes on each dimension of the $D$-dimensional array. The resulting orthogonal interactions between two dimensions produce regular grids. Typical regular-grid partition models include the infinite relational model (IRM) [17] and mixed-membership stochastic blockmodels [2]. Bayesian plaid models [24, 15, 4] generate "plaid" like partitions, however they neglect the dependence between latent feature values and more importantly, they are restricted to discrete arrays (in contrast to our continuous space model). Regular-grid partition models are widely used in real-world applications for modeling graph data [14, 33].

To our knowledge, only the Mondrian process (MP) [32, 31] and the rectangular tiling process (RTP) [25] can produce flexible axis-aligned partitions on a product space. The MP recursively generates axis-aligned cuts on a unit hypercube and partitions the space in a hierarchical fashion known as a $k$d-tree. Compared to the hierarchical partitioning strategy, the RTP generates a flat

partition structure on a two-dimensional array by assigning each entry to an existing block or a new block in sequence, without violating the rectangular restriction on the blocks.

## 2.2 Bayesian Relational Models

Most stochastic partition processes are developed to target relational modelling [17, 32, 25, 36, 9, 10]. A stochastic partition process generates a partition on a multi-dimensional array, which serves as a prior for the underlying communities in the relational data (e.g. social networks in the 2-dimensional case). After generating the partition, a local model is allocated to each block (or polygon) of the partition to characterize the relation type distribution in that block. For example, the local model can be a Bernoulli distribution for link prediction in social networks or a discrete distribution for rating prediction in recommender systems. Finally, row and column indexes are sampled to locate a block in the partition and use the local model to further generate the relational data.

Compared to the existing stochastic partition processes in relational modelling, the RBP introduces a very different partition approach: the RBP adopts a bounding-based strategy while the others are based on cutting-based strategies. This unique feature enables the RBP to directly capture important communities without wasting model parameters on unimportant regions. In addition, the bounding boxes are independently generated by the RBP. This parallel strategy is much more efficient than the hierarchical strategy [32, 9] and the entry-wise growing strategy [25].

## 2.3 Bayesian Regression Trees

The ensemble of regression/decision trees [3, 11] is a popular tool in regression tasks due to its competitive and robust performance against other models. In the Bayesian framework, Bayesian additive regression trees (BART) [5] and the Mondrian forest [18, 19] are two representative methods.

BART adopts an ensemble of trees to approximate the unknown function from the input space to the output label. Through prior regularization, BART can keep the effects of individual trees small and work as a "weak learner" in the boosting literature. BART has shown promise in nonparametric regression; and several variants of BART have been developed to focus on different scenarios, including heterogeneous BART [29] allowing for various observational variance in the space, parallel BART [30] enabling parallel computing for BART, and Dirichlet additive regression trees [22] imposing a sparse Dirichlet prior on the dimensions to address issues of high-dimensionality.

The Mondrian forest (MF) is built on the idea of an ensemble of Mondrian processes ($k$d-trees) to partition the space. The MF is distinct from BART in that the MF may be more suitable for streaming data scenarios, as the distribution of trees sampled from the MP stays invariant even if the data domain changes over time.

## 3 The Rectangular Bounding Process

The goal of the Rectangular Bounding Process (RBP) is to partition the space by attaching rectangular bounding boxes to significant regions, where "significance" is application dependent. For a hypercubical domain $X \subset \mathbb{R}^D$ with $L^{(d)}$ denoting the length of the $d$-th dimension of $X$, a budget parameter $\tau \in \mathbb{R}^+$ is used to control the expected number of generated bounding boxes in $X$ and a length parameter $\lambda \in \mathbb{R}^+$ is used to control the expected size of the generated bounding boxes. The generative process of the RBP is defined as follows:

1. Sample the number of bounding boxes $K_\tau \sim \text{Poisson}(\tau \prod_{d=1}^{D} \left[ 1 + \lambda L^{(d)} \right])$;

2. For $k = 1, \ldots, K_\tau$, $d = 1, \ldots, D$, sample the initial position $s_k^{(d)}$ and the length $l_k^{(d)}$ of the $k$-th bounding box (denoted as $\square_k$) in the $d$-th dimension:

   (a) Sample the initial position $s_k^{(d)}$ as

$$
\begin{cases}
s_k^{(d)} = 0, & \text{with probability } \frac{1}{1 + \lambda L^{(d)}}; \\
s_k^{(d)} \sim \text{Uniform}(0, L^{(d)}], & \text{with probability } \frac{\lambda L^{(d)}}{1 + \lambda L^{(d)}};
\end{cases}
$$

(b) Sample the length $l_k^{(d)}$ as

$$
\begin{cases}
l_k^{(d)} = L^{(d)} - s_k^{(d)}, & \text{with probability } e^{-\lambda(L^{(d)} - s_k^{(d)})}; \\
l_k^{(d)} \sim \text{Trun-Exp}(\lambda, L^{(d)} - s_k^{(d)}), & \text{with probability } 1 - e^{-\lambda(L^{(d)} - s_k^{(d)})}.
\end{cases}
$$

3. Sample $K_\tau$ *i.i.d.* time points uniformly in $(0, \tau]$ and index them to satisfy $t_1 < \ldots < t_{K_\tau}$. Set the cost of $\square_k$ as $m_k = t_k - t_{k-1}$ ($t_0 = 0$).

Here Trun-Exp$(\lambda, L^{(d)} - s_k^{(d)})$ refers to an exponential distribution with rate $\lambda$, truncated at $L^{(d)} - s_k^{(d)}$. The RBP is defined in a measurable space $(\Omega_X, \mathcal{B}_X)$, where $X \in \mathcal{F}(\mathbb{R}^D)$ denotes the domain and $\mathcal{F}(\mathbb{R}^D)$ denotes the collection of all finite boxes in $\mathbb{R}^D$. Each element in $\Omega_X$ denotes a partition $\boxplus_X$ of $X$, comprising a collection of rectangular bounding boxes $\{\square_k\}_k$, where $k \in \mathbb{N}$ indexes the bounding boxes in $\boxplus_X$. A bounding box is defined by an outer product $\square_k := \bigotimes_{d=1}^{D} u_k^{(d)}([0, L^{(d)}])$, where $u_k^{(d)}$ is a step function defined on $[0, L^{(d)}]$, taking value of "1" in $[s_k^{(d)}, s_k^{(d)} + l_k^{(d)}]$ and "0" otherwise (see right panel of Figure 1).

Given a domain $X$, hyperparameters $\tau$ and $\lambda$, a random partition sampled from the RBP can be represented as: $\boxplus_X \sim \text{RBP}(X, \tau, \lambda)$. We assume that the costs of bounding boxes are *i.i.d.* sampled from the same exponential distribution, which implies there exists a homogeneous Poisson process on the time (cost) line. The generating time of each bounding box is uniform in $(0, \tau]$ and the number of bounding boxes has a Poisson distribution. We represent a random partition as $\boxplus_X := \{\square_k, m_k\}_{k=1}^{K_\tau} \in \Omega_X$.

## 3.1 Expected Total Volume

**Proposition 1.** *Given a hypercubical domain $X \subset \mathbb{R}^D$ with $L^{(d)}$ denoting the length of the $d$-th dimension of $X$ and the value of $\tau$, the expected total volume of the bounding boxes (i.e. expected number of boxes × expected volume of a box) in $\boxplus_X$ sampled from a RBP is a constant $\tau \cdot \prod_{d=1}^{D} L^{(d)}$.*

The expected length of the interval in $u_k^{(d)}$ with value "1" is $\mathbb{E}(|u_k^{(d)}|) = \mathbb{E}(l_k^{(d)}) = \frac{L^{(d)}}{1 + \lambda L^{(d)}}$. According to the definition of the RBP (Poisson distribution in Step 1), we have an expected number of $\tau \cdot \prod_{d=1}^{D} [1 + \lambda L^{(d)}]$ bounding boxes in $\boxplus_X$. Thus, the expected total volume of the bounding boxes for a given budget $\tau$ and a domain $X$ is $\tau \cdot \prod_{d=1}^{D} L^{(d)}$.

Proposition 1 implies that, given $\tau$ and $X$, the RBP generates either many small-sized bounding boxes or few large-sized bounding boxes. This provides a practical guidance on how to choose appropriate values of $\lambda$ and $\tau$ when implementing the RBP. Given the lengths $\{L^{(d)}\}_d$ of $X$, an estimate of the lengths of the bounding boxes can help to choose $\lambda$ (i.e. $\lambda = \frac{L^{(d)} - \mathbb{E}(|u_k^{(d)}|)}{L^{(d)} \mathbb{E}(|u_k^{(d)}|)}$). An appropriate value of $\tau$ can then be chosen to determine the expected number of bounding boxes.

## 3.2 Coverage Probability

**Proposition 2.** *For any data point $x \in X$ (including the boundaries of $X$), the probability of $x^{(d)}$ falling in the interval of $[s_k^{(d)}, s_k^{(d)} + l_k^{(d)}]$ on $u_k^{(d)}$ is a constant $\frac{1}{1 + \lambda L^{(d)}}$ (and does not depend on $x^{(d)}$). As the step functions $\{u_k^{(d)}\}_k$ for constructing the $k$-th bounding box $\square_k$ in $\boxplus_X$ are independent, $x$ is covered by $\square_k$ with a constant probability.*

The property of constant coverage probability is particularly suitable for regression problems. Proposition 2 implies there is no biased tendency to favour different data regions in $X$. All data points have equal probability to be covered by a bounding box in a RBP partition $\boxplus_X$.

Another interesting observation can be seen from this property: Although we have specified a direction for generating the $d$-th dimension of $\square_k$ in the generative process (i.e. the initial position $s_k^{(d)}$ and the terminal position $v_k^{(d)} = s_k^{(d)} + l_k^{(d)}$), the probability of generating $u^{(d)}$ is the same if we reverse the direction of the $d$-th dimension, which is $p(s_k^{(d)}, v_k^{(d)}) = \frac{e^{-\lambda l_k^{(d)}}}{1 + \lambda L^{(d)}} \cdot \lambda^{\mathbf{1}_{s_k^{(d)} > 0} + \mathbf{1}_{v_k^{(d)} < L}}$. It is obvious that the joint probability of the initial position and the terminal position is invariant

if we reverse the direction of the $d$-th dimension. Direction is therefore only defined for notation convenience – it will not affect the results of any analysis.

### 3.3 Self-Consistency

The RBP has the attractive property of self-consistency. That is, while restricting an RBP$(Y, \tau, \lambda)$ on $D$-dimensional hypercube $Y$, to a sub-domain $X$, $X \subset Y \in \mathcal{F}(\mathbb{R}^D)$, the resulting bounding boxes restricted to $X$ are distributed as if they are directly generated on $X$ through an RBP$(X, \tau, \lambda)$ (see Figure 2 for an illustration). Typical application scenarios are regression/classification tasks on streaming data, where new data points may be observed outside the current data domain $X$, say falling in $Y/X, X \subset Y$, where $Y$ represents the data domain (minimum bounding box of all observed data) in the future (left panel of Figure 3). Equipped with the self-consistency property, the distribution of the RBP partition on $X$ remains invariant as new data points come and expand the data domain.

The self-consistency property can be verified in three steps: (1) the distribution of the number of bounding boxes is self-consistent; (2) the position distribution of a bounding box is self-consistent; (3) the RBP is self-consistent. In the following, we use $\pi_{Y,X}$ to denote the projection that restricts $\boxplus_Y \in \Omega_Y$ to $X$ by keeping $\boxplus_Y$'s partition in $X$ unchanged and removing the rest.

**Proposition 3.** *While restricting the RBP$(Y, \tau, \lambda)$ to $X$, $X \subset Y \in \mathcal{F}(\mathbb{R}^D)$, we have the following results:*

1. *The time points of bounding boxes crossing into $X$ from $Y$ follow the same Poisson process for generating the time points of bounding boxes in a RBP$(X, \tau, \lambda)$. That is*
   $$P^Y_{K_\tau, \{m_k\}_{k=1}^{K_\tau}} \left( \pi_{Y,X}^{-1} \left( K_\tau^X, \{m_k^X\}_{k=1}^{K_\tau^X} \right) \right) = P^X_{K_\tau, \{m_k\}_{k=1}^{K_\tau}} \left( K_\tau^X, \{m_k^X\}_{k=1}^{K_\tau^X} \right).$$

2. *The marginal probability of the pre-images of a bounding box $\square^X$ in $Y$ (given the bounding box in $Y$ would cross into $X$) equals the probability of $\square^X$ being directly sampled from a RBP$(X, \tau, \lambda)$. That is, $P^Y_\square \left( \pi_{Y,X}^{-1}(\square^X) \mid \pi_{Y,X}(\square^Y) \neq \emptyset \right) = P^X_\square(\square^X)$.*

3. *Combining 1 and 2 leads to the self-consistency of the RBP: $P^Y_\boxplus \left( \pi_{Y,X}^{-1}(\boxplus_X) \right) = P^X_\boxplus(\boxplus_X)$.*

The generative process provides a way of defining the RBP on a multidimensional finite space (hypercube). According to the Kolmogorov extension theorem ([6]), we can further extend RBP to the multidimensional infinite space $\mathbb{R}^D$.

**Theorem 1.** *The probability measure $P^X_\boxplus$ on the measurable space $(\Omega_X, \mathcal{B}_X)$ of the RBP, $X \in \mathcal{F}(\mathbb{R}^D)$, can be uniquely extended to $P^{\mathbb{R}^D}_\boxplus$ on $(\Omega_{\mathbb{R}^D}, \mathcal{B}_{\mathbb{R}^D})$ as the projective limit measurable space.*

## 4 Application to Regression Trees

### 4.1 RBP-Regression Trees

We apply the RBP as a space partition prior to the regression-tree problem. Given the feature and label pairs $\{(x_n, y_n)\}_{n=1}^N, (x_n, y_n) \in \mathbb{R}^D \times \mathbb{R}$, the bounding boxes $\{\square_k\}_k$ sampled from an RBP on the domain $X$ of the feature data are used for modelling the latent variables that influence the observed labels. Since the $\{\square_k\}_k$ allow for a partition layout of overlapped bounding boxes, each single feature point can be covered by more than one bounding box, whose latent variables together form an additive effect to influence the corresponding label.

The generative process for the RBP-regression tree is as follows:

$$\{\square_k\}_k \sim \text{RBP}(X, \tau, \lambda); \quad \omega_k \sim \mathcal{N}(\mu_\omega, \epsilon_\omega^2); \quad \sigma^2 \sim \mathcal{IG}(1, 1); \quad y_n \sim \mathcal{N}(\sum_k \omega_k \cdot \mathbf{1}_{x \in \square_k}(x_n), \sigma^2).$$

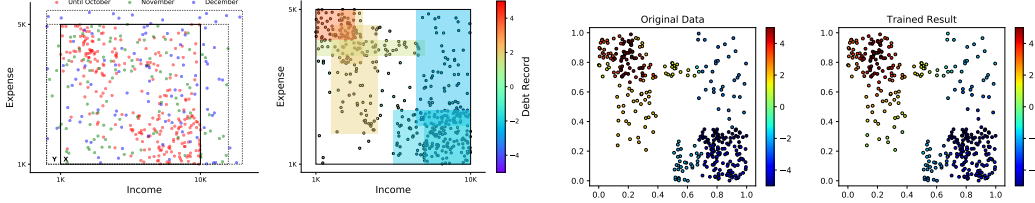

Figure 2: Self-consistency of the Rectangular Bounding Process: $P^Y_{\boxplus}\left(\pi^{-1}_{Y,X}(\boxplus_X)\right) = P^X_{\boxplus}(\boxplus_X)$.

Figure 3: A toy regression-tree example: (left) Debt $\sim f(\text{Income}, \text{Expense})$, for some some probability density function, $f$. When entering November from October, more observed records exceeding Income of $\$10K$ or Expense of $\$5K$ are observed, where $X \subset Y$ denotes the data domain in October and $Y$ the data domain in November. (right) The RBP regression-tree predicts the original data well, particularly in dense and complex regions (top left and bottom right).

## 4.2 Sampling for RBP-Regression Tree

The joint probability of the label data $\{y_n\}_{n=1}^N$, the number of bounding boxes $K_\tau$, the variables related to the bounding boxes $\{\omega_k, s_k^{(1:D)}, l_k^{(1:D)}\}_{k=1}^{K_\tau}$, and the error variance $\sigma^2$ is

$$P(\{y_n\}_n, K_\tau, \{\omega_k, s_k^{(1:D)}, l_k^{(1:D)}\}_k, \sigma^2 | \lambda, \tau, \mu_\omega, \epsilon_\omega^2, X, L^{(1:D)}) = \prod_n P(y_n | K_\tau, \{\omega_k, s_k^{(1:D)}, l_k^{(1:D)}\}_k, \sigma^2, X)$$

$$\cdot P(K_\tau | \tau, \lambda, L^{(1:D)}) K_\tau! P(\sigma^2) \prod_k P(\omega_k | \mu_\omega, \epsilon_\omega^2) \cdot \prod_{k,d} P(s_k^{(d)} | \lambda, L^{(d)}) P(l_k^{(d)} | s_k^{(d)}, \lambda, L^{(d)})$$

We adopt MCMC methods to iteratively sampling from the resulting posterior distribution.

**Sample $K_\tau$:** We use a similar strategy to [1] for updating $K_\tau$. We accept the addition or removal of a bounding box with an acceptance probability of $\min(1, \alpha_{\text{add}})$ or $\min(1, \alpha_{\text{del}})$ respectively, where

$$\alpha_{\text{add}} = \frac{\prod_n P_{K_\tau+1}(y_n) \cdot \tau\lambda_*(1-P_0)}{\prod_n P_{K_\tau}(y_n) \cdot (K_\tau+1)P_0}, \quad \alpha_{\text{del}} = \frac{\prod_n P_{K_\tau-1}(y_n) \cdot K_\tau P_0}{\prod_n P_{K_\tau}(y_n) \cdot \tau\lambda_*(1-P_0)},$$

$\lambda_* = \prod_d \left[1 + \lambda L^{(d)}\right]$, $P_{K_\tau}(y_n) = P(y_n | K_\tau, \{\omega_k, s_k^{(d)}, l_k^{(d)}\}_k, \sigma^2, \{x_n\}_n)$, and $P_0 = \frac{1}{2}$ (or $1 - P_0$) denotes the probability of proposing to add (or remove) a bounding box.

**Sample $\sigma^2, \{\omega_k\}_k$:** Both $\sigma^2$ and $\{\omega_k\}_k$ can be sampled through Gibbs sampling:

$$\sigma^2 \sim \mathcal{IG}\left(1 + \frac{N}{2}, 1 + \frac{\sum_n(y_n - \sum_k \omega_k \cdot \mathbf{1}_{x \in \square_k}(x_n))^2}{2}\right), \quad \omega_k^* \sim \mathcal{N}(\mu^*, (\sigma^2)^*),$$

where $\mu^* = (\sigma^2)^*\left(\frac{\mu_\omega}{\epsilon_\omega^2} + \frac{\sum_n\left(y_n - \sum_{k' \neq k} \omega_{k'} \cdot \mathbf{1}_{x \in \square_{k'}}(x_n)\right)}{\sigma^2}\right)$, $(\sigma^2)^* = \left(\frac{1}{\epsilon_\omega^2} + \frac{N_k}{\sigma^2}\right)^{-1}$, and $N_k$ is the number of data points belonging to the $k$-th bounding box.

**Sample $\{s_k^{(d)}, l_k^{(d)}\}_{k,d}$:** To update $u_k^{(d)}$ we use the Metropolis-Hastings algorithm, generating the proposed $s_k^{(d)}, l_k^{(d)}$ (which determine $u_k^{(d)}$) using the RBP generative process (Step 2.(a)(b)). Thus, the acceptance probability is based purely on the likelihoods for the proposed and current configurations.

## 4.3 Experiments

We empirically test the RBP regression tree (RBP-RT) for the regression task. We compare the RBP-RT with several state-of-the-art methods: (1) a Bayesian additive regression tree (BART) [5];

Table 1: Regression task comparison results (RMAE±std)

| Data Sets | RF | ERT | BART | MF | RBP-RT |
|---|---|---|---|---|---|
| Protein | $3.20 \pm 0.03$ | $\mathbf{3.04} \pm 0.03$ | $4.50 \pm 0.05$ | $4.79 \pm 0.07$ | $4.87 \pm 0.10$ |
| Naval Plants | $0.35 \pm 0.01$ | $\mathbf{0.13} \pm 0.01$ | $0.46 \pm 0.07$ | $0.37 \pm 0.10$ | $0.53 \pm 0.17$ |
| Power Plants | $4.03 \pm 0.08$ | $\mathbf{3.65} \pm 0.08$ | $4.36 \pm 0.11$ | $5.03 \pm 0.12$ | $4.78 \pm 0.17$ |
| Concrete Data | $4.18 \pm 0.31$ | $\mathbf{3.80} \pm 0.33$ | $5.95 \pm 0.56$ | $6.21 \pm 0.46$ | $6.34 \pm 0.78$ |
| Airfoil self-noise | $1.06 \pm 0.06$ | $\mathbf{0.74} \pm 0.08$ | $1.21 \pm 0.24$ | $1.10 \pm 0.21$ | $1.25 \pm 0.33$ |

(2) a Mondrian forest (MF) [18, 19]; (3) a random forest (RF) [3]; and (4) an extremely randomized tree (ERT) [12]. For BART, we implement a particle MCMC strategy to infer the structure of each tree in the model; for the MF, we directly use the existing code provided by the author[1]; for the RF and ERT, we use the implementations in the scikit-learn toolbox [28].

**Synthetic data:** We first test the RBP-RT on a simple synthetic data set. A total of 7 bounding boxes are assigned to the unit square $[0, 1]^2$, each with its own mean intensity $\omega_k$. From each bounding box $50 \sim 80$ points are uniformly sampled, with the label data $y_i$ generated from a normal distribution, with mean the sum of the intensities of the bounding boxes covering the data point, and standard deviation $\sigma = 0.1$. In this way, a total of 400 data points are generated in $[0, 1]^2$.

To implement the RBP-TR we set the total number of iterations to 500, $\lambda = 2$ (i.e. the expected box length is $\frac{1}{3}$) and $\tau = 1$ (i.e. the expected number of bounding boxes is 90). It is worth noting that 90 is a relatively small number of blocks when compared to the other state-of-the-art methods. For instance, BART usually sets the number of trees to be at least 50 and there are typically more than 16 blocks in each tree (i.e. at least 800 blocks in total). The right panel of Figure 3 shows a visualization of the data fitting result of the RBP regression-tree.

**Real-world data:** We select several real-world data sets to compare the RBP-RT and the other state-of-the-art methods: Protein Structure [8] ($N = 45,730, D = 9$), Naval Plants [7] ($N = 11,934, D = 16$), Power Plants [34] ($N = 9,569, D = 4$), Concrete [37] ($N = 1,030, D = 8$), and Airfoil Self-Noise [8] ($N = 1,503, D = 8$). Here, we first use PCA to select the 4 largest components and then normalize them so that they lie in the unit hypercube for ease of implementation. As before, we set the total number of iterations to 500, $\lambda = 2$ and this time set $\tau = 2$ (i.e. the expected number of bounding boxes is 180).

The resulting Residual Mean Absolute Errors (RMAE) are reported in Table 1. In general, the three Bayesian tree ensembles perform worse than the random forests. This may in part be due to the maturity of development of the RF algorithm toolbox. While the RBP-RT does not perform as well as the random forests, its performance is comparable to that of BART and MF (sometimes even better), but with many fewer bounding boxes (i.e. parameters) used in the model, clearly demonstrating its parsimonious construction.

## 5 Application to Relational Modeling

### 5.1 The RBP-Relational Model

Another possible application of the RBP is in relational modeling. Given relational data as an asymmetric matrix $R \in \{0, 1\}^{N \times N}$, with $R_{ij}$ indicating the relation from node $i$ to node $j$, the bounding boxes $\{\square_k\}_k$ with rates $\{\omega_k\}_k$ belonging to a partition $\boxplus$ may be used to model communities with different intensities of relations.

The generative process of an RBP relational model is as follows: (1) Generate a partition $\boxplus$ on $[0, 1]^2$; (2) for $k = 1, \ldots, K_\tau$, generate rates $\omega_k \sim \text{Exp}(1)$; (3) for $i, j = 1, \ldots, N$, generate the row and column coordinates $\{\xi_i\}_i, \{\eta_j\}_j$; (4) for $i, j = 1, \ldots, N$, generate relational data $R_{ij} \sim \text{Bernoulli}(\sigma(\sum_{k=1}^{K_\tau} \omega_k \cdot \mathbf{1}_{(\xi, \eta) \in \square_k}(\xi_i, \eta_j)))$, where $\sigma(x) = \frac{1}{1 + \exp(-x)}$ is the logistic function, mapping the aggregated relation intensity from $\mathbb{R}$ to $(0, 1)$. While here we implement a RBP relational model with binary interactions (i.e. the Bernoulli likelihood), other types of relations (e.g. categorical likelihoods) can easily be accommodated.

Table 2: Relational modeling (link prediction) comparison results (AUC±std)

| Data Sets | IRM | LFRM | MP-RM | BSP-RM | MTA-RM | RBP-RM |
|---|---|---|---|---|---|---|
| Digg | $0.80 \pm 0.01$ | $0.81 \pm 0.03$ | $0.79 \pm 0.02$ | $\mathbf{0.82} \pm 0.02$ | $0.83 \pm 0.01$ | $\mathbf{0.83} \pm 0.01$ |
| Flickr | $0.88 \pm 0.01$ | $0.89 \pm 0.01$ | $0.88 \pm 0.01$ | $\mathbf{0.93} \pm 0.02$ | $0.90 \pm 0.01$ | $\mathbf{0.92} \pm 0.01$ |
| Gplus | $0.86 \pm 0.01$ | $0.86 \pm 0.01$ | $0.86 \pm 0.01$ | $\mathbf{0.89} \pm 0.02$ | $0.86 \pm 0.01$ | $\mathbf{0.88} \pm 0.01$ |
| Facebook | $0.87 \pm 0.01$ | $0.91 \pm 0.02$ | $0.89 \pm 0.03$ | $\mathbf{0.93} \pm 0.02$ | $0.91 \pm 0.01$ | $\mathbf{0.92} \pm 0.02$ |
| Twitter | $0.87 \pm 0.01$ | $0.88 \pm 0.02$ | $0.88 \pm 0.06$ | $\mathbf{0.90} \pm 0.01$ | $0.88 \pm 0.01$ | $\mathbf{0.90} \pm 0.02$ |

Together, the RBP and the mapping function $\sigma(\cdot)$ play the role of the random function $W(\cdot)$ defined in the graphon literature [27]. Along with the uniformly generated coordinates for each node, the RBP relational model is expected to uncover homogeneous interactions in $R$ as compact boxes.

## 5.2 Sampling for RBP-Relational Model

The joint probability of the label data $\{R_{ij}\}_{i,j}$, the number of bounding boxes $K_\tau$, the variables related to the bounding boxes $\{\omega_k, s_k^{(1:D)}, l_k^{(1:D)}\}_{k=1}^{K_\tau}$, and the coordinates $\{\xi_n, \eta_n\}_n$ for the nodes (with $L^{(1)} = \ldots = L^{(D)} = 1$ in the RBP relational model) is

$$P(R, K_\tau, \{\omega_k, s_k^{(1:D)}, l_k^{(1:D)}\}_k, \{\xi_n, \eta_n\}_n | \lambda, \tau) = \prod_{n_1, n_2} P(R_{n_1,n_2} | K_\tau, \{\omega_k, s_k^{(1:D)}, l_k^{(1:D)}\}_k, \xi_{n_1}, \eta_{n_2})$$

$$\cdot P(K_\tau | \tau, \lambda) K_\tau! \prod_{n_1} P(\xi_{n_1}) \prod_{n_2} P(\eta_{n_2}) \prod_k P(\omega_k) \prod_{k,d} P(s_k^{(d)} | \lambda) P(l_k^{(d)} | s_k^{(d)}, \lambda).$$

We adopt MCMC methods to iteratively sample from the resulting posterior distribution.

**Sample $K_\tau$:** We use a similar strategy to [1] for updating $K_\tau$. We accept the addition or removal of a bounding box with an acceptance probability of $\min(1, \alpha_{\text{add}})$ or $\min(1, \alpha_{\text{del}})$ respectively, where

$$\alpha_{\text{add}} = \frac{\prod_{n_1, n_2} P_{K_\tau+1}(R_{n_1,n_2}) \cdot \tau \lambda_*(1 - P_0)}{\prod_{n_1, n_2} P_{K_\tau}(R_{n_1,n_2}) \cdot (K_\tau + 1) P_0}, \quad \alpha_{\text{del}} = \frac{\prod_{n_1, n_2} P_{K_\tau-1}(R_{n_1,n_2}) \cdot K_\tau P_0}{\prod_{n_1, n_2} P_{K_\tau}(R_{n_1,n_2}) \cdot \tau \lambda_*(1 - P_0)},$$

where $\lambda_* = (1 + \lambda)^2$, $P_{K_\tau}(R_{n_1,n_2}) = P(R_{n_1,n_2} | K_\tau, \{\omega_k, s_k^{(1:D)}, l_k^{(1:D)}\}_k, \xi_{n_1}, \eta_{n_2})$ and $P_0 = \frac{1}{2}$ (or $1 - P_0$) denotes the probability of proposing to add (or remove) a bounding box.

**Sample $\{\omega_k\}_k$:** For the $k$-th box, $k \in \{1, \cdots, K_\tau\}$, a new $\omega_k^*$ is sampled from the proposal distribution $\text{Exp}(1)$. We then accept $\omega_k^*$ with probability $\min(1, \alpha)$, where

$$\alpha = \prod_{n_1, n_2} \frac{P(R_{n_1,n_2} | K_\tau, \{\omega_{k'}\}_{k' \neq k}, \omega_k^*, \{s_{k'}^{(1:D)}, l_{k'}^{(1:D)}\}_{k'}, \xi_{n_1}, \eta_{n_2})}{P(R_{n_1,n_2} | K_\tau, \{\omega_k, s_k^{(1:D)}, l_k^{(1:D)}\}_k, \xi_{n_1}, \eta_{n_2})}.$$

**Sample $\{u_k^{(d)}\}_{k,d}$:** This update is the same as for the RBP-regression tree sampler (Section 4.2).

**Sample $\{\xi_{n_1}\}_{n_1}, \{\eta_{n_2}\}_{n_2}$** We propose new $\xi_{n_1}, \eta_{n_2}$ values from the uniform prior distribution. Thus, the acceptance probability is purely based on the likelihoods of the proposed and current configurations.

## 5.3 Experiments

We empirically test the RBP relational model (RBP-RM) for link prediction. We compare the RBP-RM with four state-of-the-art relational models: (1) IRM [17] (regular grids); (2) LFRM [24] (plaid grids); (3) MP relational model (MP-RM) [32] (hierarchical $k$d-tree); (4) BSP-tree relational model (BSP-RM) [9]; (5) Matrix tile analysis relational model (MTA-RM) [13] (noncontiguous tiles). For inference on the IRM and LFRM, we adopt collapsed Gibbs sampling algorithms, for MP-RM we use reversible-jump MCMC [35], for BSP-RM we use particle MCMC, and for MTA-RM we implement the iterative conditional modes algorithm used in [13].

**Data Sets**: Five social network data sets are used: Digg, Flickr [38], Gplus [23], Facebook and Twitter [20]. We extract a subset of nodes (the top 1000 active nodes based on their interactions with others) from each data set for constructing the relational data matrix.

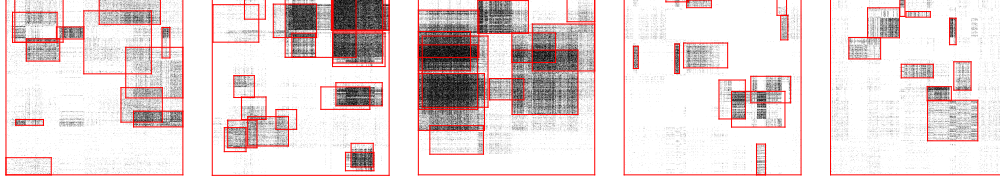

Figure 4: Visualisation of the RBP relational model partition structure for five relational data sets: (left to right) Digg, Flickr, Gplus, Facebook and Twitter.

**Experimental Setting**: The hyper-parameters for each method are set as follows: In IRM, we let the concentration parameter $\alpha$ be sampled from a gamma $\Gamma(1,1)$ prior, and the row and column partitions be sampled from two independent Dirichlet processes; In LFRM, we let $\alpha$ be sampled from a gamma $\Gamma(2,1)$ prior. As the budget parameter for MP-RM and BSP-RM is hard to sample [19], we set it to 3, implying that around $(3+1) \times (3+1)$ blocks would be generated. For the parametric model MTA-RM, we simply set the number of tiles to 16; In RBP-RM, we set $\lambda = 0.99$ and $\tau = 3$, which leads to an expectation of 12 boxes. The reported performance is averaged over 10 randomly selected hold-out test sets (Train:Test = 9:1).

**Results**: Table 2 reports the link prediction performance comparisons for each method and datasets. We see that the RBP-RM achieves competitive performance against the other methods. Even on the data sets it does not obtain the best score, its performance is comparable to the best. The overall results validate that the RBP-RM is effective in relational modelling due to its flexible and parsimonious construction, attaching bounding boxes to dense data regions.

Figure 4 visually illustrates the RBP-RM partitions patterns for each dataset. As is apparent, the bounding-based RBP-RM method indeed describing dense regions of relational data matrices with relatively few bounding boxes (i.e. parameters). An interesting observation from this partition format, is that the overlapping bounding boxes are very useful for describing inter-community interactions (e.g. overlapping bounding boxes in Digg, Flickr, and Gplus) and community-in-community interactions (e.g. upper-right corner in Flickr and Gplus). Thus, in addition to competitive and parsimonious performance, the RBP-RM also produces intuitively interpretable and meaningful partitions (Figure 4).

## 6 Conclusion

We have introduced a novel and parsimonious stochastic partition process – the Rectangular Bounding Process (RBP). Instead of the typical cutting-based strategy of existing partition models, we adopt a bounding-based strategy to attach rectangular bounding boxes to model dense data regions in a multi-dimensional space, which is able to avoid unnecessary dissections in sparse data regions. The RBP was demonstrated in two applications: regression trees and relational modelling. The experimental results on these two applications validate the merit of the RBP, that is, competitive performance against existing state-of-the-art methods, while using fewer bounding boxes (i.e. fewer parameters).

## Acknowledgements

Xuhui Fan and Scott A. Sisson are supported by the Australian Research Council through the Australian Centre of Excellence in Mathematical and Statistical Frontiers (ACEMS, CE140100049), and Scott A. Sisson through the Discovery Project Scheme (DP160102544). Bin Li is supported by Fudan University Startup Research Grant and Shanghai Municipal Science & Technology Commission (16JC1420401).

## Footnotes

[1]https://github.com/balajiln/mondrianforest

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
