[Supplementary Material]

# Rectangular Bounding Process: Supplementary Material

**Xuhui Fan**
School of Mathematics & Statistics
University of New South Wales
xuhui.fan@unsw.edu.au

**Bin Li**
School of Computer Science
Fudan University
libin@fudan.edu.cn

**Scott A. Sisson**
School of Mathematics & Statistics
University of New South Wales
scott.sisson@unsw.edu.au

## 1 Calculation of the expected side length in Proposition 1

The expected length of the interval in $u_k^{(d)}$ with value "1" is (some super/subscripts are omitted in this proof for conciseness):

$$
\begin{aligned}
\mathbb{E}(l) \;\overset{(1)}{=}\;& P(s=0)\mathbb{E}(l|s=0) + \int_0^L p(s)\mathbb{E}(l|s)ds \\
=\;& \frac{1}{1+\lambda L}\left[\int_0^L l\lambda e^{-\lambda l}dl + Le^{-\lambda L}\right] + \frac{\lambda}{1+\lambda L}\cdot\int_0^L\left[\int_0^{L-s} l\lambda e^{-\lambda l}dl + (L-s)e^{-\lambda(L-s)}\right]ds \\
=\;& \frac{1}{1+\lambda L}\left[\lambda\cdot(-\frac{1}{\lambda^2}-\frac{l}{\lambda})e^{-\lambda l}|_0^L + Le^{-\lambda L}\right] \\
& + \frac{\lambda}{1+\lambda L}\cdot\int_0^L\left[\lambda\cdot(-\frac{1}{\lambda^2}-\frac{l}{\lambda})e^{-\lambda l}|_0^{L-s} + (L-s)e^{-\lambda(L-s)}\right]ds \\
=\;& \frac{L}{1+\lambda L}
\end{aligned}
\tag{1}
$$

where the first term after $\overset{(1)}{=}$ refers to the expected length of the interval when starting at $0$ and the second term refers to the expected length when starting from a point larger than $0$. We need to use the equality $\frac{d\left[(-\frac{1}{\lambda^2}-\frac{x}{\lambda})e^{-\lambda x}\right]}{dx} = xe^{-\lambda x}$ in the above derivation.

## 2 Proof for Proposition 2 (Coverage Probability)

For any data point $x \in X$ (including the boundaries of $X$), the probability of $x^{(d)}$ falling in the interval of $[s_k^{(d)}, s_k^{(d)}+l_k^{(d)}]$ on $u_k^{(d)}$ is a constant $\frac{1}{1+\lambda L^{(d)}}$ (some super/subscripts are omitted in this proof for conciseness).

If $x^{(d)}$ locates on the initial boundary, $u^{(d)}(0) = 1$ *i.f.f.* $s^{(d)} = 0$, which is

$$
P(u^{(d)}(0) = 1) = \frac{1}{1+\lambda L}
\tag{2}
$$

If $x^{(d)} > 0$, we have the corresponding probability as

$$
\begin{aligned}
P(u^{(d)}(x) = 1) &= P(s=0)P(l > x) + \int_0^x p(s)P(l > x - s)ds \qquad (3)\\
&= \frac{1}{1+\lambda L}e^{-\lambda x} + \frac{\lambda e^{-\lambda x}}{1+\lambda L} \cdot \int_0^x e^{\lambda s}ds = \frac{e^{-\lambda x}}{1+\lambda L}\left(1 + e^{\lambda s}|_0^x\right) = \frac{1}{1+\lambda L}
\end{aligned}
$$

## 3 Proof for Proposition 3 (Self-Consistency)

**Proposition 3.1: The number distribution of bounding boxes is self-consistent**

*Proof.* According to the definition of Poisson process, the bounding boxes sampled from $\mathrm{RBP}(Y, \tau, \lambda)$ (or $\mathrm{RBP}(X, \tau, \lambda)$) follow a homogeneous Poisson process with intensity $\prod_d (1 + \lambda L_Y^{(d)})$ (or $\prod_d (1 + \lambda L_X^{(d)})$). Given the same budget $\tau$, the result holds if we can prove the following equality of the two Poisson process intensities

$$
\prod_d (1 + \lambda L_Y^{(d)}) \cdot P\left(\pi_{Y,X}(\square^Y) \neq \emptyset\right) = \prod_d (1 + \lambda L_X^{(d)}) \qquad (4)
$$

Due to the independence of dimensions, $P\left(\pi_{Y,X}(\square^Y) \neq \emptyset\right)$ can be rewritten as

$$
P\left(\pi_{Y,X}(\square^Y) \neq \emptyset\right) = \prod_d P(|\pi_{Y,X}(u_Y^{(d)})| > 0) \qquad (5)
$$

where we use $|\pi_{Y,X}(u_Y^{(d)})| > 0$ to denote the case that the step function $u_Y^{(d)}$ would take value "1" in an interval of the $d$-th dimension of domain $X$.

W.l.o.g, we assume that the two domains, $X$ and $Y$, have the same shape apart from the $d'$-th dimension where the length of dimension $L_Y^{(d')}$ in $Y$ is larger than that of in $X$.

There are three cases to consider: (1) $X$ and $Y$ share the terminal boundary in the $d'$-th dimension; (2) $X$ and $Y$ share the initial boundary in the $d'$-th dimension; (3) $X$ and $Y$ do not share the boundaries in the $d'$-th dimension. Proving case (1) and case (2) together would obviously lead to case (3). In either case, by independence of dimensions, we need to prove the following equation.

$$
(1 + \lambda L_Y^{(d')}) \cdot P\left(|\pi_{Y,X}(u_Y^{(d')})| > 0\right) = (1 + \lambda L_X^{(d')}) \qquad (6)
$$

In case (1) where $X$ and $Y$ share the terminal boundary in the $d'$th dimension, we have

$$
\begin{aligned}
&P\left(|\pi_{Y,X}(u_Y^{(d')})| > 0\right) \qquad (7)\\
&\overset{(2)}{=} P(s_Y^{(d')} \in (L_Y^{(d')} - L_X^{(d')}, L_Y^{(d')}]) + P(s_Y^{(d')} = 0)P(l_Y^{(d')} > (L_Y^{(d')} - L_X^{(d')})|s_Y^{(d')} = 0)\\
&\quad + \int_{0+}^{L_Y^{(d')} - L_X^{(d')}} P(s_Y^{(d')})P(l_Y^{(d')} > L_Y^{(d')} - L_X^{(d')} - s_Y^{(d')}|s_Y^{(d')})ds_Y^{(d')}\\
&= \frac{\lambda L_X^{(d')}}{1 + \lambda L_Y^{(d')}} + \frac{1}{1 + \lambda L_Y^{(d')}} \cdot e^{-\lambda(L_Y^{(d')} - L_X^{(d')})}\\
&\quad + \int_{0+}^{L_Y^{(d')} - L_X^{(d')}} \frac{\lambda}{1 + \lambda L_Y^{(d')}} \cdot e^{-\lambda(L_Y^{(d')} - L_X^{(d')} - s_Y^{(d')})}ds_Y^{(d')}\\
&= \frac{1 + \lambda L_X^{(d')}}{1 + \lambda L_Y^{(d')}}
\end{aligned}
$$

where the first term after $\overset{(2)}{=}$ corresponds to the probability that $s_Y^{(d')}$ locates directly in the interval of $(L_Y^{(d')} - L_X^{(d')}, L_Y^{(d')}]$, the second term corresponds to the probability that $s_Y^{(d')}$ locates on the initial boundary and has the length larger than $L_Y^{(d')} - L_X^{(d')}$, and the third term corresponds to the

probability that $s_Y^{(d')}$ locates in the interval of $(0, L_Y^{(d')} - L_X^{(d')}]$ (excluding the initial boundary) and has the length larger than $L_Y^{(d')} - L_X^{(d')} - s_Y^{(d')}$.

In case (2) where $X$ and $Y$ share the initial boundary in the $d'$th dimension, we have

$$P\left(|\pi_{Y,X}(u_Y^{(d')})| > 0\right) = \frac{1 + \lambda L_X^{(d')}}{1 + \lambda L_Y^{(d')}} \tag{8}$$

since $|\pi_{Y,X}(u_Y^{(d')})| > 0$ requires the condition of $s_Y^{(d')} \in [0, L_X^{(d')}]$ because $s_Y^{(d')} \notin [0, L_X^{(d')}]$ would lead to the result that $\pi_{Y,X}(u_Y^{(d')}) = 0$.

The conclusion can be derived as above. □

Because of the same Poisson process intensity Eq. (4), the following equality also holds

$$P_{K_\tau, \{m_k\}_{k=1}^{K_\tau}}^Y \left(\pi_{Y,X}^{-1}\left(K_\tau^X, \{m_k^X\}_{k=1}^{K_\tau^X}\right)\right) = P_{K_\tau, \{m_k\}_{k=1}^{K_\tau}}^X \left(K_\tau^X, \{m_k^X\}_{k=1}^{K_\tau^X}\right) \tag{9}$$

**Proposition 3.2: The position distribution of a bounding box is self-consistent**

*Proof.* W.l.o.g, we assume that the two domains, $X$ and $Y$, have the same shape apart from the $d'$th dimension where $Y$ has additional space of length $L'$ ($L' > 0$) (the general case follows by induction). For dimensions $d \neq d'$, it is obvious that the law of bounding boxes are consistent under projection because the projection is the identity. Given the same budget $\tau$, The result holds if we can prove the following equality

$$P_u^Y \left(\pi_{Y,X}^{-1}(u_X^{(d')}) \mid |\pi_{Y,X}(u_Y^{(d')})| > 0\right) = P_u^X(u_X^{(d')}), \tag{10}$$

There are two cases to consider: (1) $X$ and $Y$ share the initial boundary in the $d'$th dimension; (2) $X$ and $Y$ share the terminal boundary in the $d'$th dimension. In each case, there are two cases (denoted as A & B in the following) regarding whether the terminal/initial (for Case 1/2, respectively) position locates on the boundary of $X$. In total we have four cases to discuss as follows.

In case (1) where $X$ and $Y$ share the initial boundary, according to Eq. (8), we have $P\left(|\pi_{Y,X}(u_Y^{(d)})| > 0\right) = \frac{1+\lambda L_X^{(d')}}{1+\lambda L_Y^{(d')}}$.

For convenience of notation, we let $\theta_\dagger = P_s^Y(s_Y^{(d')} \mid s_Y^{(d')} \in X)$, specifically $\theta_\dagger = \frac{1}{1+\lambda L_X^{(d')}}$ if $s_Y^{(d')} = 0$; $\theta_\dagger = \frac{\lambda}{1+\lambda L_X^{(d')}}$ if $s_Y^{(d')} > 0$.

[Case 1.A] For $0 < s_X^{(d')} + l_X^{(d')} < L_X^{(d')} < L_Y^{(d')}$,

$$P_u^Y \left(\pi_{Y,X}^{-1}(u_X^{(d')}) \mid |\pi_{Y,X}(u_Y^{(d')})| > 0\right) = \theta_\dagger \cdot \lambda e^{-\lambda l_X^{(d')}} = P_u^X(u_X^{(d')}). \tag{11}$$

[Case 1.B] For $0 < s_X^{(d')} + l_X^{(d')} = L_X^{(d')} < L_Y^{(d')}$,

$$\begin{aligned}&P_u^Y \left(\pi_{Y,X}^{-1}(u_X^{(d')}) \mid |\pi_{Y,X}(u_Y^{(d')})| > 0\right)\\ &= P(s_Y^{(d')})(P(l_Y^{(d')} > (L_X^{(d')} - s_Y^{(d')}))) = \theta_\dagger e^{-\lambda(L_X^{(d')} - s_Y^{(d')})} = P_u^X(u_X^{(d')})\end{aligned} \tag{12}$$

In case (2) where $X$ and $Y$ share the terminal boundary, according to Eq. (7), we have $P\left(|\pi_{Y,X}(u_Y^{(d)})| > 0\right) = \frac{1+\lambda L_X^{(d')}}{1+\lambda L_Y^{(d')}}$.

[Case 2.A] For $\pi_{Y,X}(s_Y^{(d')}) = 0$, we have $s_X^{(d')} = 0$. What is more, we have

$$P\left(\pi_{Y,X}(s_Y^{(d')}) = 0\right) \tag{13}$$
$$= P(s_Y^{(d')} = 0)P\left(l_Y^{(d')} > (L_Y^{(d')} - L_X^{(d')})|s_Y^{(d')} = 0\right)$$
$$+ \int_{0+}^{L_Y^{(d')} - L_X^{(d')}} P(s^{(d')})P(l > (L_Y^{(d')} - L_X^{(d')} - s^{(d')})|s^{(d')})ds^{(d')}$$
$$= \frac{1}{1 + \lambda L_Y^{(d')}}$$

Thus we can get

$$P_u^Y\left(\pi_{Y,X}^{-1}(u_X^{(d')}) \mid |\pi_{Y,X}(u_Y^{(d')})| > 0\right) \tag{14}$$
$$= P(\pi_{Y,X}(S_Y^{(d')}) = 0)P(l_X^{(d')})/P(|\pi_{Y,X}(u_Y^{(d')})| > 0)$$
$$= \frac{1}{1 + L_X^{(d')}} \cdot \theta_{\ddagger} = P_u^X(u_X^{(d')}),$$

where $\theta_{\ddagger} = e^{-\lambda l_X^{(d')}}$ if $\pi_{Y,X}(s_Y^{(d')}) + l_X^{(d')} = L_X^{(d')}$; $\theta_{\ddagger} = \lambda e^{-\lambda l_X^{(d')}}$ if $\pi_{Y,X}(s_Y^{(d')}) + l_X^{(d')} < L_X^{(d')}$.

[Case 2.B] For $\pi_{Y,X}(s_Y^{(d')}) > 0$, we have $s_X^{(d')} > 0$,

$$P_u^Y\left(\pi_{Y,X}^{-1}(u_X^{(d')}) \mid |\pi_{Y,X}(u_Y^{(d')})| > 0\right) \tag{15}$$
$$= P(\pi_{Y,X}(S_Y^{(d')}) = s)P(l_X^{(d')})/P(|\pi_{Y,X}(u_Y^{(d')})| > 0)$$
$$= \frac{\lambda}{1 + L_X^{(d')}} \cdot \theta_{\ddagger} = P_u^X(u_X^{(d')}),$$

Consider all $D$ dimensions, for each case, we have $P_{\Box}^Y\left(\pi_{Y,X}^{-1}(\Box^X) \mid \pi_{Y,X}(\Box^Y) \neq \emptyset\right) = P_{\Box}^X(\Box^X)$.
□

### Proposition 3.3: RBP is self-consistent

$$P_{\boxplus}^Y(\pi_{Y,X}^{-1}(\boxplus_X)) = P_{\boxplus}^Y\left(\pi_{Y,X}^{-1}\left(K_{\tau}^X, \{m_k^X, \Box_k^X\}_{k=1}^{K_{\tau}^X}\right)\right)$$

$$= P_{K_{\tau}, \{m_k\}_k}^Y\left(\pi_{Y,X}^{-1}\left(K_{\tau}^X, \{m_k^X\}_{k=1}^{K_{\tau}^X}\right)\right) \cdot P_{\Box}^Y\left(\pi_{Y,X}^{-1}\left(\{\Box_k^X\}_{k=1}^{K_{\tau}^X}|K_{\tau}^X, \{m_k^X\}_{k=1}^{K_{\tau}^X}\right)\right) \tag{16}$$

$$= P_{K_{\tau}, \{m_k\}_k}^Y\left(\pi_{Y,X}^{-1}\left(K_{\tau}^X, \{m_k^X\}_{k=1}^{K_{\tau}^X}\right)\right) \cdot P_{\Box}^Y\left(\pi_{Y,X}^{-1}\left(\{\Box_k^X\}_{k=1}^{K_{\tau}^X}|K_{\tau}^X\right)\right) \tag{17}$$

$$= P_{K_{\tau}, \{m_k\}_k}^Y\left(\pi_{Y,X}^{-1}\left(K_{\tau}^X, \{m_k^X\}_{k=1}^{K_{\tau}^X}\right)\right) \cdot \prod_{k=1}^{K_{\tau}^X} P_{\Box}^Y\left(\pi_{Y,X}^{-1}\left(\Box_k^X\right)\right) \tag{18}$$

$$= P_{K_{\tau}, \{m_k\}_k}^X\left(K_{\tau}^X, \{m_k^X\}_{k=1}^{K_{\tau}^X}\right) \cdot \prod_{k=1}^{K_{\tau}^X} P_{\Box}^Y\left(\pi_{Y,X}^{-1}\left(\Box_k^X\right)\right) \tag{19}$$

$$= P_{K_{\tau}, \{m_k\}_k}^X\left(K_{\tau}^X, \{m_k^X\}_{k=1}^{K_{\tau}^X}\right) \cdot \prod_{k=1}^{K_{\tau}^X} P_{\Box}^X\left(\Box_k^X\right) \tag{20}$$

$$= P_{\boxplus}^X\left(K_{\tau}^X, \{m_k^X, \Box_k^X\}_{k=1}^{K_{\tau}^X}\right) = P_{\boxplus}^X(\boxplus_X).$$

We can obtain Eq. (17) from Eq. (16) because

$$P\left(\{m_k, \Box_k\}_{k=1}^{K_{\tau}}|K_{\tau}\right) = P\left(\{m_k\}_{k=1}^{K_{\tau}}|K_{\tau}\right) \cdot P\left(\{\Box_k\}_{k=1}^{K_{\tau}}|K_{\tau}\right)$$

which indicates

$$P\left(\{\Box_k\}_{k=1}^{K_\tau}|K_\tau, \{m_k\}_{k=1}^{K_\tau}\right) = P\left(\{\Box_k\}_{k=1}^{K_\tau}|K_\tau\right)$$

We can obtain Eq. (18) from Eq. (17) because of independence of bounding boxes. Eq. (19) is derived from Eq. (18) by applying Proposition 3.1 while Eq. (20) is derived from Eq. (19) by applying Proposition 3.2.