[Reviews · NeurIPS 2018]

Reviewer 1



The paper describes a process to generate (grid-aligned) boxes in spaces of arbitrary dimension. The authors show the process obeys many desirable/necessary properties in the construction of Bayesian nonparametric models. In some regards, I would have fought for the paper to be accepted: I like how simply the model is described and motivated, and the experiments are wonderfully broad and thorough. But the paper makes serious errors on omitting necessary technical components that a paper such as this should have. For example, the model is briefly described in Sec. 3, but nowhere is a likelihood function (joint distribution over the model) provided! Neither are any descriptions of what the posterior distributions look like (or which ones we're interested in targeting), which leads me to my biggest criticism of the paper: The serious misjudgment of the authors to leave any details of inference out of the paper. I believe the only cases where inference details can be relegated to the supplementary material is when the algorithm used is a direct, unaltered application of an existing algorithm, and even then the paper should contain a high-level (technical) description. This is especially troubling for this paper since, as the authors themselves point out, this partitioning model is fundamentally different from those previously presented in the literature (this idea of bounding rather than cutting). Moreover, while reading the paper, I was very interested in the construction of the model, and in my head my main thought was "the most interesting part will be seeing how inference is performed." For example, sampling the start positions and lengths of each interval during the construction is beautifully simple, but when you have data in the grid, and given the position of one interval (on one axis), what does the updated distribution on the interval of a second axis look like? How would it be sampled? These are the most crucial (and meaningful) questions to ask when thinking about Bayesian approaches to inference. An even more sour taste is left in my mouth because the authors decided to instead keep Sec. 3.1--3.3 in the paper. Sure these are interesting results but it is unforgivable to have devoted so much to their description at the sacrifice of details of your inference procedure. I did read the details of inference in the supp (though I shouldn't have to) and it would have been far more useful to have that in the paper and even expanded, arduously. This is a shame, your work has all the elements of being a great NIPS paper, and I would have fought for it. It really just comes down to what you chose to (and not to) write and how you've organized. I'll discuss with the AE. More detailed comments: p. 2, line 42: "budget" is never defined. p. 2, line 69 "...restricted in discrete arrays.": I don't know what this means. p. 3, line 94: I wouldn't say the ensembles of the trees are "easy-to-explain". Can you explain what a random forest learns? p. 3, line 119: How do you sample P( l_k^{(d)} )? p. 2 & 3: model description: Unless I missed it, I don't see what the "costs" are for. I am assuming that they somehow play a role in the likelihood and thus in the inference procedure... Sec. 3.2 coverage probability: I'm confused... in what cases or types of construction would the coverage probability depend on the datapoint? IMO, it would be a very unnatural construction if it did. Sec. 5 Experiments: This is great work. Very thorough and broad and beyond what I normally see in NIPS papers. I like the two demonstrative applications to relational modeling and regression trees. Some additional comments: p. 6, Paragraph starting line 217: You really should put quick details of these datasets in the text. Such as size and some characterizing properties. p. 8, paragraph starting line 259: This paragraph makes no sense to me. Read it a few times I still have no idea what it's saying. p. 8, par starting line 267: What is \alpha? Small complaint: I don't like the self-descriptions of your model of "parsimonious" or "comprehensive". For example, in the abstract and intro and demonstrably p. 8, line 286. Can't you just say "visually intuitive (given evidence such as in Fig. 5)" or something like that? Updated review: My original score of 3 was a strong (and accurate) statement, voicing my opinions about what was not included in the paper, making it incomplete and inappropriate for acceptance. After reading the authors' response, I believe they agree with my assessment and will rectify this in the final version of the paper. If that is indeed corrected, then the paper is a good submission, justifying a score of 7 (in my opinion), which I stated in my original review. I will check back in on the final version and raise an objection to the AE and editors if this is these promises are not kept.

Reviewer 2



I have now read all the reviews and the authors' rebuttal. I will stick to my original opinion and happy to have this paper accepted. I fell confident that the authors will address the main concerns (mainly re presentation and inclusion of important details) raised by the reviewers. -------------------------------------------------------------------------------- The authors propose a stochastic partition process, the Rectangular Bounding Process (RBP). Motivated by the unnecessary cuts dictated by the cut-based approaches, the RBP is a bounded-based approach that fits bounding boxes to regions in an informed manner based on their significance (dense data regions). I found the proposed approach interesting. It definitely feels that solves pathogenies observed in previous methodologies and adds only an incremental contribution. However, the model is neat and technically sound – to my understanding. The authors provide experiments where the process is applied on relational data allowing for the deployment of the process in interesting domains. The performance is not outstanding compared to other methodologies, however this does not degrade it’s contribution as a mathematical neat and interesting process. The work reads easily and is clear through and through. I suggest acceptance. Other: Line 109: BSP --> RBP Line 220: lies --> lie

Reviewer 3



The authors define a new random partition of product spaces, called Rectangular Bounding Process (RBP). Stochastic partition models are useful to divide a product space into a certain number of regions, in such a way that data in each region are \textit{homogeneous}. A well-known an example of these types of models is the Mondrian process defined in Roy and Teh (2009). The authors state that many models in existing literature produce unnecessary cuts in regions where data are rare, and the RBP tries to overcome such an issue. In Section 2 the authors review some previous works on the subject, in Section 3 they define the RBP, describing some of its properties, among which the self--consistency. At the end of the paper they provide some applications (regression trees and relational modeling). I think that the paper is not sufficiently clear and not well written, furthermore the definition of the RBP is not rigorous and precise. There are many obscure points that are not appropriately explained and discussed, and too many inaccuracies throughout the manuscript. I suggest to reject the paper, and below I list the main points which support my conclusion and help the authors to rewrite the whole paper for possible publication elsewhere. 1) In the introduction the authors have to underline and explain why stochastic partition model are important tools in real applications. 2) Section 2 is not well organized, it seems to me a long list of previous contributions and related works without a clear focus. The authors should rewrite the section in a more systematic way, limiting themselves to underline the difference between the RBP with respect to the other existing stochastic partition processes. 3) Is there a big difference in terms of flexibility between the Mondrian process and the proposed RBP? What does the RBP bring with respect to the Mondrian process? 4) Pg. 2, line 42. What does it mean ``expected total volume''? Please, specify the definition. 5) Pg. 3, line 110. What does it mean ``significant regions''? How do you define a significant region? 6) Pg. 3, line 112-113. $\lambda, \tau$ are introduced here, the authors say that they are parameters, do they belong to $\R$ or to a subset of $\R$? The domain must be specified. 7) Pg. 3, line 118. Is $P (s_k^{(d)})$ a probability distribution? It seems that this is not a probability distribution, indeed \begin{align*} P(s_{k}^{(d)} \in [0, L^{(d)}]) &= P(s_{k}^{(d)}=0)+P(s_{k}^{(d)} \in (0, L^{(d)}]) = \frac{1}{1+\lambda L^{(d)}} + \frac{\lambda}{1+\lambda L^{(d)}}\int_0^{L^{(d)}}\frac{1}{L^{(d)}} \D s\& = \frac{1+\lambda}{1+\lambda L^{(d)}} \not = 1 \end{align*} Furthermore the notation $P (s_k^{(d)})$ has not actual probabilistic meaning, one should use $\Lcr(s_{k}^{(d)})$ to denote the law of the random variable, otherwise $P(s_{k}^{(d)} \in A)$ to denote the probability that the random variable $s_{k}^{(d)}$ belongs to the Borel set $A$. This comment applies throughout the manuscript. 8) Pg. 3, line 121. The role of the cost of $\square_k$ is not clear. 9) Pg. 4, line 122. What does it mean ``The generative process defines a measurable space''? This sentence has no actual meaning, a random variable cannot define a measurable space. 10) Pg. 4, line 122. $\mathcal{F}(\R^D)$ has not been defined. 11) Pg.4, line 123. ``Each element in $\Omega_X$ denotes a partition $\boxplus_X$'' a partition of what? Probably a partition of $X$. 12) Pg. 4, line 125. $\square_k$ was a box, now is a product of functions. What do the authors mean with the notation $\bigotimes_{d=1}^D u_k^{(d)} ([0, L^{(d)}])$? It seems to me a product of indicator functions, but the authors speak of an outer product. I think that they have to clarify this point. 13) Pg. 4, Proposition 1. What do you mean with the phrase ``the value of $\tau$''? In the proof of Proposition 1 (see the supplementary material) the authors evaluate $\E[l_k^{(d)}]$, which is not the expected total volume of the bounding boxes mentioned in the statement of the proposition. The final step of the proof is given just after the statement of Proposition 1. This is very confusing. 14) Pg. 5, Figure 3. This figure has to be removed and the self-consistency has to be explained by words. Some minor comments are listed below: 1) Pg. 1, line 26. Replace ``As'' with ``as''; 2) Pg. 3, line 121. The notation $\square_k$ is not elegant; 3) Pg. 4, line 124. Teplace ``boxs'' with ``boxes''; 4) Pg. 5, line 183. Teplace ``Combining1'' with ``Combining 1''. References. D.M. Roy and Y.W. Teh (2009). The Mondrian process. In \textit{NIPS}, pp. 1377--1384